# CRISPR/Cas9-Based Mutagenesis of Starch Biosynthetic Genes in Sweet Potato (Ipomoea Batatas) for the Improvement of Starch Quality

**DOI:** 10.3390/ijms20194702

**Published:** 2019-09-23

**Authors:** Hongxia Wang, Yinliang Wu, Yandi Zhang, Jun Yang, Weijuan Fan, Hui Zhang, Shanshan Zhao, Ling Yuan, Peng Zhang

**Affiliations:** 1National Key Laboratory of Plant Molecular Genetics, CAS Center for Excellence in Molecular Plant Sciences, Institute of Plant Physiology and Ecology, Shanghai Institutes for Biological Sciences, Chinese Academy of Science, Shanghai 200032, China; HXWANG@SIBS.AC.CN (H.W.); ylwu01@sibs.ac.cn (Y.W.); ydzhang2016@sibs.ac.cn (Y.Z.); 2Department of Plant and Soil Sciences and Kentucky Tobacco Research and Development Center, University of Kentucky, Lexington, KY 40546, USA; 3Shanghai Sanshu Biotechnology Co., LTD. Shanghai 201210, China; 4University of Chinese Academy of Sciences, Beijing 100049, China; 5Shanghai Key Laboratory of Plant Functional Genomics and Resources, Shanghai Chenshan Plant Science Research Center, Chinese Academy of Science, Shanghai Chenshan Botanical Garden, Shanghai 201602, China; jyang03@sibs.ac.cn (J.Y.); wjfan@sibs.ac.cn (W.F.); sszhao@sibs.ac.cn (S.Z.); 6Shanghai Center for Plant Stress Biology, Chinese Academy of Sciences, Shanghai 201602, China; huizhang@sibs.ac.cn

**Keywords:** CRISPR/Cas9, Genome editing, *IbGBSSI*, *IbSBEII*, sweet potato

## Abstract

CRISPR/Cas9-mediated genome editing is a powerful technology that has been used for the genetic modification of a number of crop species. In order to evaluate the efficacy of CRISPR/Cas9 technology in the root crop, sweet potato (*Ipomoea batatas*), two starch biosynthetic pathway genes, *IbGBSSI* (encoding granule-bound starch synthase I), and *IbSBEII* (encoding starch branching enzyme II), were targeted in the starch-type cultivar Xushu22 and carotenoid-rich cultivar Taizhong6. *I. batatas* was transformed using a binary vector, in which the *Cas9* gene is driven by the *Arabidopsis* AtUBQ promoter and the guide RNA is controlled by the *Arabidopsis* AtU6 promoter. A total of 72 Xushu22 and 35 Taizhong6 transgenic lines were generated and analyzed for mutations. The mutation efficiency was 62–92% with multi-allelic mutations in both cultivars. Most of the mutations were nucleotide substitutions that lead to amino acid changes and, less frequently, stop codons. In addition, short nucleotide insertions or deletions were also found in both *IbGBSSI* and *IbSBEII*. Furthermore, a 2658 bp deletion was found in one *IbSBEII* transgenic line. The total starch contents were not significantly changed in *IbGBSSI*- and *IbSBEII*-knockout transgenic lines compared to the wild-type control. However, in the allopolyploid sweet potato, the *IbGBSSI*-knockout reduced, while the *IbSBEII*-knockout increased, the amylose percentage. Our results demonstrate that CRISPR/Cas9 technology is an effective tool for the improvement of starch qualities in sweet potato and breeding of polyploid root crops.

## 1. Introduction

Genome-editing mediated by CRISPR/Cas9 is a revolutionary technology that enables the loss or gain of function of specific genes to produce desired qualities in plants [1,2]. The CRISPR/Cas9-knockout technique is relatively simple to perform with high accuracy [3,4]. The CRISPR/Cas9 system has been successfully used in plant breeding of crops with valuable traits, such as antioxidant-rich purple tomatoes, omega-3 fatty acid-enriched oil crop, starch altered potato, and high-yielding rice [5,6,7,8,9,10]. As removal of the transgenes in CRISPR/Cas9-edited plants is a prerequisite for gaining regulatory approval for commercial applications, T-DNA-free genome editing can potentially speed up the breeding process in agriculture [11,12].

Sweet potato (*Ipomoea batatas*), is one of the top starch-rich root crops worldwide, as well as a source for human nutrition and industrial raw materials [13]. The biosynthesis of starch requires five classes of core enzymes, including ADP-glucose pyrophosphorylase (ADPG), starch synthases (SS), starch branching enzymes (SBEs), starch debranching enzymes (DBEs), and granule-bound starch synthase I (GBSSI), that are located in the chloroplast or amyloplast [14,15,16]. GBSSI is responsible for amylose biosynthesis, and SBEs, including starch branching enzyme I (SBEI) and starch branching enzyme II (SBEII), are involved in amylopectin biosynthesis [17]. Suppression of *GBSS* by co-suppression or RNA interference (RNAi) results in the production of amylose-free starch [18,19,20]. Shimada et al. successfully obtained a higher amylose content by RNA interference of the *IbSBEII* gene [21]. Compared to SBEI, SBEII generates a higher rate of branching due to an apparent higher affinity towards amylopectin for the transferring of shorter glucan chains [22]. Nevertheless, starch biosynthesis in sweet potato is still insufficiently understood. Compared to many other major crops, such as rice, maize, soybean, and cassava, the functional genomic study of the sweet potato has been difficult due to the low efficiency of gene transformation, allopolyploidy (2*n* = 6*x* = 90), and limited genomic information. The recent achievements of gene transformation and genome mapping of sweet potato have opened the door to studying gene functions in this important crop using CRISPR/Cas9-mediated genome-editing [23,24,25].

In this study, targeted mutagenesis of two starch biosynthetic pathway genes, *IbGBSSI* and *IbSBEII*, were achieved by using the CRISPR/Cas9 system in two sweet potato cultivars, starch-type cultivar Xushu22 and carotenoid-rich cultivar Taizhong6. Our results provide the first demonstration of CRISPR-Cas9-mediated genome editing of the sweet potato, confirming that multi-copy gene knockout in polyploid plants is achievable with phenotypic consequences.

## 2. Results

### 2.1. Single or Dual gRNA Vectors Allow Evaluation of Efficiency for Targeted Mutagenesis

CRISPR/Cas9-based vectors were constructed for the transformation of *I. batatas*. We designed the gRNAs using online gRNA design tools and the available *I. batatas* draft genome sequences [24]. To evaluate the editing efficiency, two vectors, harboring either a single gRNA or a double gRNA cassette with the Cas9 endonuclease, were constructed and used for transformation (Figure 1a). The U6 promoter from *Arabidopsis* (pAtU6), was used to drive the gRNA expression. The *Arabidopsis* ubiquitin-1 promoter (AtUBQ1) was used to control Cas9 expression. The *IbGBSSI* gene contains nine exons. Two vectors, one containing a single gRNA (sgRNA2) and the other having double gRNAs (sgRNA12: sgRNA2 + sgRNA1), were used to target the first exon (Figure 1b). For the *IbSBEII* gene knockout, the single gRNA vector (IbSBEII-sgRNA2) aimed to mutate exon 15, and the dual gRNA vector IbSBEII-sgRNA12 (IbSBEII-sgRNA2 + IbSBEII-sgRNA1) were used to target exons 12 and 15 (Figure 1c, Appendix A). *I. batatas* is self-incompatible, we thus chose two cultivars, Xushu22 and Taizhong6, for the transformation and subsequent removal of CRISPR/Cas9 T-DNA by crossing. The gene-knockout cultivars thus generated will be used as parental lines in future breeding programs to obtain desirable starch quality.

### 2.2. Efficient Gene Mutagenesis in Sweet Potato

The *I. batatas* genome is comprised of two B_1_ and four B_2_ component genomes (B_1_B_1_B_2_B_2_B_2_B_2_) [26]. A multi-step scheme was deployed to mutate the target genes (Figure 2). First, the four vectors were transformed into Xushu22 and Taizhong6 via *Agrobacterium*-mediated method and putative transgenic plants were generated. PCR analysis using genomic DNA isolated from the transgenic lines confirmed the stable integration of the T-DNA into the genome. Next, we used gene-specific primers, more than 200 bp flanking the target sites, in a series of combinatorial PCR using genomic DNA isolated from the transgenic lines. In the events in which significant deletions or insertions occur, band size shift might appear in gel electrophoresis. In most cases, point mutations and small deletions/insertions were detected by DNA sequencing of the PCR products. We identified 107 transgenic sweet potato lines using PCR. We subsequently sequenced the gene-specific PCR fragments, flanking the target sites, amplified using DNA isolated from the putative transgenic lines. Of the 25 IbGBSSI-sgRNA transgenic lines of Xushu22, 23 contained mutations, a 92% mutation rate. Of the 24 IbGBSSI-sgRNA transgenic lines of Taizhong6, 15 contained mutations in the target gene, a 62% mutation rate. In total, 40 of the 47 IbSBEII transgenic lines of Xushu22 contained gene-specific mutations, including 24 of the 25 double sgRNA transgenic lines and 16 of the 22 single sgRNA transgenic lines. Additionally, 11 transgenic IbSBEII-sgRNA lines were obtained for Taizhong6, of which 63% had mutations in the target gene. The overall mutation rates of Xushu22 appeared to be higher than those of Taizhong6 (Table 1). Vectors with double gRNAs also appeared to generate higher mutation rates compared with those containing single gRNA. Sequencing of the PCR products of *IbGBSSI* and *IbSBEII* genes from the transgenic lines revealed various mutations, including nucleotide deletion, insertion, and substitution. In Xushu22, six of the eight IbGBSSI-sgRNA2 transgenic lines contained mutations, of which one line had a short deletion, four lines had single point mutations, and one line had a combination of point mutation and deletion (Appendix A). In contrast, all 17 IbGBSSI-sgRNA12 transgenic lines had single nucleotide insertions at the +89 bp position. Moreover, sequence analysis showed that IbGBSSI-sgRNA12-1 and IbGBSSI-sgRNA12-2 lines have 12 bp and 15 bp deletions close to the PAM sequence, respectively. Further, 24 of the 25 transgenic lines of IbSBEII-sgRNA12 contained mutations, as did 16 of the 22 IbSBEII-sgRNA2 transgenic lines. It thus appears that vectors with two gRNA (IbSBEII-sgRNA12) have higher mutation efficiency than that with a single gRNA. As a typical example, IbSBEII-sgRNA2-21 in Xushu22 produced one PCR band with similar size as the wild-type (WT), and DNA sequencing revealed the presence of nucleotide substitutions, insertion (A-TC) and a short deletion (TTTACTGGCTTTAAGCAGCCTA) (Figure 3, Appendix A). Similarly, IbSBEII-sgRNA12-24 produced a single small PCR band by PCR amplification. Sequencing of small bands just revealed a major peak that contains a 2658 bp deletion between two of the sgRNA in Xushu22 (Figure 3b; Appendix A). IbSBEII-sgRNA12-4 generated a mutation at +6811 bp (CC to GG), a 6 bp deletion (TGATAA) at +6802 bp in exon 12, and a single nucleotide (T) insertion at +6484 bp and an 8 bp deletion (AAGCAGCC) at +9475 bp in exon 15.

In Taizhong6, of the 15 IbGBSSI-sgRNA2 transgenic lines, two lines had insertions, five lines had single point mutations, one line combined a single site mutation and an insertion, and another line had both a point mutation and a deletion (Appendix A). The same mutation, AAC to GTT, occurred in IbGBSSI-sgRNA12 lines 3, 5, 6, and 7. A single point mutation and a deletion occurred in IbGBSSI-sgRNA12-10 transgenic line. In IbGBSSI-sgRNA2 transgenic lines, six lines had single point mutations and one line had both a single point mutation and a deletion. In IbSBEII-sgRNA12 transgenic lines, two lines had short deletions (Appendix A). Sequencing results revealed that IbSBEII-sgRNA12-6 has a 6 bp deletion from 6826–6831 bp (GTTATC) and an 11 bp deletion (GGGTTATCATT) from 6829–6839 bp. IbSBEII-sgRNA2-12 and IbSBEII-sgRNA2-15 have 1 bp deletions that result in frame shifts. Collectively, our findings suggest that PCR combined with sequencing can be used to analyze the precision of CRISPR/Cas9-targeted mutagenesis in the polyploid sweet potato.

### 2.3. The Starch and Amylose Contents and Chain Length Distribution in Mutated Plants

The starch and amylose contents were quantified in the storage roots of IbGBSSI-sgRNA and IbSBEII-sgRNA transgenic lines of Xushu22 and Taizhong6 (Figure 4). The starch contents of all transgenic lines were not significantly changed compared to WT (Figure 4a). However, because IbGBSSI controls amylose biosynthesis, relative proportions of amylose in IbGBSSI-knockout transgenic lines were significantly reduced compared to WT (Figure 4b). In Xushu22 knockout lines, amylose contents ranged from 5.75% (IbGBSSI-sgRNA12-7) to 22.4% (IbGBSSI-sgRNA2-5). Low amylose contents were also achieved in Taizhong6 knockout lines, ranging from 5.50% to 14.8%. SBEII is involved in amylopectin biosynthesis; knocking out *SBEII* led to the decrease of amylopectin and increase of amylose. The proportions of amylose in IbSBEII-sgRNA lines were higher compared to WT. The highest increase (38–40.3%) of amylose was found in lines IbSBEII-sgRNA12-26 and IbSBEII-sgRNA12-23 compared to Xushu22 WT (27.2%). In the Taizhong6 background, amylose contents reached 37.8% in IbSBEII-sgRNA2-15 and 37.4% in IbSBEII-sgRNA2-16 compared to 25.1% in WT. IbSBEII-sgRNA2-20 and IbSBEII-sgRNA2-7 were confirmed as transgenic lines by PCR amplification of the Cas9 gene. Sequence analysis did not detect mutations; however, the amylose contents were significantly different than that of WT. We speculated that a minor mutation peak in the mixed PCR products was masked by the major peak.

To further identify the physicochemical properties of starches in transgenic lines, the amylopectin chain length distribution was determined in the short chain of 90 > DP > 6 and the long chain of DP >43 according to peak area. In Xushu22 background, significant differences were detected in short (DP 6–12) (Figure 5a) and long chains (DP ≥200) (Figure 5b) when the profiles of Xushu22, IbSBEII-sgRNA and IbGBSSI-sgRNA were compared with each other. Chain length distribution in DP 6–12 reached 16.4% in IbSBEII-sgRNA12-26 and 19.4% in IbSBEII-sgRNA2-1 compared to 29.3% in Xushu22 and 30.5%–35.0% in IbGBSSI-sgRNA2 (Table 2). Therefore, the high-amylose starch in IbSBEII-sgRNAs had fewer short chains and more long chains compared to WT and IbGBSSI-sgRNAs.

## 3. Discussion

Compared with other genome editing technologies, such as zinc finger nucleases (ZFNs) and transcription activator-like effector nucleases (TALENs), the CRISPR/Cas9 system is simple to use and cost-effective while having high mutagenic efficiency. It has been successfully used in a wide range of plant species, including some allopolyploid crops (e.g., wheat, potato, soybean, rapeseed, and cotton) [27,28,29,30,31,32]. However, an example has yet to be available for sweet potato. In this study, we successfully achieved the first CRISPR/Cas9-meditated mutagenesis in two sweet potato cultivars and produced higher amylose and amylopectin for improving the starch quality in sweet potato. A 62–92% mutation efficiency was observed in this study, suggesting the efficient expression of gRNA in sweet potato driven by the AtU6 promoter and using the AtUBQ promoter to control Cas9 expression seems to be highly effective. CRISPR/Cas9-mediated gene knockout with high mutation rates have been reported in polyploid cotton (48–82%) and rapeseed (28–98%) [31,32]. Most of the mutations were short nucleotide substitutions, insertions or deletions; a longer segment deletion was detected in one line (Appendix A). Similar to what has been shown in other works [33,34], vectors harboring two gRNAs seem to generate higher mutation rates in *I. batatas* than those that only contain one gRNA. The analysis of targeted mutagenesis is difficult in a hexaploid genome. Ideally, a gRNA recognizes a short region, adjacent to the PAM sequence, that contains a restriction enzyme cutting site. However, we were unable to identify restriction enzyme sites adjacent to the PAM sequences with the target genes, and thus chose to use a PCR/sequencing approach to analyze the mutations.

The amylose/amylopectin ratio can affect the physicochemical properties of starches used as raw materials [35]. Starches with very low amylose (<10%) tend to be waxy [36]. High-amylose starch has higher viscosity, while high-amylopectin starch has lower gelatinization and retrogradation temperatures [37]. In tuber crop potato, high-amylose starch has been produced by CRISPR/Cas9-targeted SBEII knockout, and in cassava, RNAi knockdown of *GBSSI* has resulted in waxy starch [10,38]. Therefore, targeting biosynthetic pathway genes of starch can effectively alter starch functionality. Starch has been improved in several crops, such as corn, rice, and wheat, for value-added utilization [39]. In our study, the amylose content reached 40.3% by targeted *SBEII* mutagenesis. High-amylose, degradation-resistant starch (RS) can be specifically used as a dietary fiber for controlling obesity and diabetes [40,41,42,43]. The high-amylopectin starch from the IbGBSSI-knockout sweet potato, with an amylose content of only 5.5%, is potentially advantageous for frozen products and polymer applications due to the suppression of retrogradation [44]. Although amylose-free and moderately high-amylose starches have been obtained by RNA interference of GBSSI and SBEII genes in sweet potato [19,20], products thus produced contain T-DNA. In our study, potentially T-DNA-free plants can produce a wide range of mutations for improvement of starch quality. Moreover, the two unique *I. batatas* cultivars, with varying compositions of starch and carotenoid, are excellent parental lines in genetic crossing to produce novel properties for food and industrial applications.

In conclusion, we have developed targeted mutagenesis in sweet potato using a highly efficient CRISPR/Cas9 system. We have demonstrated the approach to alter the amylose/amylopectin ratio through selective knockout of IbGBSSI or IbSBEII. The CRISPR/Cas9 technology will also advance gene function studies and breeding of new cultivars of sweet potato. Genome editing-assisted sweet potato breeding, which allows simultaneous modifications of multiple genetic loci involved in starch and carotenoids biosynthesis, will accelerate our efforts to improve the values of sweet potato as a nutritional food and industrial raw material.

## 4. Materials and Methods

### 4.1. Vector Construction

The CRISPR/Cas9 vector was donated by Jian-Kang Zhu lab of Yanfei Mao. The plasmid contained the *Cas9* gene (under the AtUBQ1 promoter) with a nuclear localization signal (NLS) and the *Arabidopsis* AtU6 gene promoter controlling a single guide RNA (sgRNA) gene (Figure 1A). The sgRNA scaffold clones with 20 nt target sequences were obtained by polymerase chain reaction (PCR) using a pair of synthetic primers according to the manufacturer’s instructions. This amplified fragment was inserted into the KpnI and EcoRI sites of the intermediate vector to make a construct with both of the two customized sgRNAs. This intermediate construct was further digested by EcoRI and HindIII and ligated into the pCAMBIA1300 binary vector for plant transformation.

### 4.2. Agrobacterium-Mediated Transformation of Sweet Potato

The stable transformation of sweet potato was performed using the protocol of Yang et al. [18]. Briefly, Xushu22 and Taizhong6 embryogenic calli were induced from the bud tissues in one month. Calli were then sub-cultured in Murashige and Skoog (MS) medium with 2,4-D for multiplication before being transferred to liquid cell suspension (LCP) medium for further multiplication. The embryogenic suspensions were transformed with *Agrobacterium tumefaciens* strain LB4404 harboring CRISPR/Cas9 vector and co-cultivated for three days in darkness. The transformed calli were selected on fresh MS medium containing 200 mg·l^−1^ cefotaxime to eliminate *Agrobacterium*, and 10 mg·l^−1^ hygromycin for selection. The hygromycin-resistant embryogenic calli were transferred onto plant regeneration medium to develop into plantlets.

### 4.3. Detection of the Mutation of Target Genes

Genomic DNA was isolated by using the cetyltrimethyl ammonium bromide (CTAB, Sigma, Shanghai, China) method according to the manufacturer’s protocol. To detect mutations of target genes in transgenic sweet potato, primer pairs were designed at a distance of approximately 200 bp from the PAM site for precise sequencing of the target genes (Appendix A). The target fragments were amplified using the genomic DNA as a template and purified for sequencing. The mutation rate in transgenic plants were measured by directly sequencing the PCR products. Comparing PCR products separated by gel electrophoresis to the target genes revealed that the PCR products included insertions, deletions, or substitutions.

### 4.4. Detection of Total Starch and Amylose Content in the Mutants

The total starch was isolated and determined from the storage roots of sweet potato plants as described by Wang et al. [45]. Storage roots (100 mg) were ground to extract starch. In brief, the sample was suspended in 0.2 mL of aqueous ethanol (80% *v*/*v*) and shaken on a vortex. After this step, 3 mL of thermostable α-amylase solution (100 U/mL) was added to degrade the starch into sugar, and then samples were boiled for 6 min and vigorously shaken three times. The final volume was diluted into 10 mL with distilled water. Starch content was measured by using the Megazyme kit (Megazyme International Ireland Ltd. Co., Wicklow, Ireland). Changes in sugar content were measured by using a spectrophotometer at a wavelength of 510 nm and mathematically converting the measurements into starch content. The total starch content was counted from the glucose content according to the following formula: starch content (mg/g FW) = glucose content × 162/180 (adjustment from free D-glucose to anhydro-D-glucose, as occurs in starch). The amylose ratio was then quantified by using the pure potato amylose and amylopectin as a standard sample (Sigma, Shanghai, China). Amylose content was measured using a spectrophotometer at a wavelength of 710 nm.

### 4.5. Detection of Chain Length Distribution

According to Zhou et al. and Zhang et al., the chain length distributions of amylopectin were quantitatively analyzed using high-performance anion-exchange chromatography with pulsed amperometric detection (HPAEC-PAD; Dionex-ICS 3000; Dionex Corporation, Sunnyvale, CA, USA) for DP6-90, and the Agilent 1100 Series SEC system was used with GRAM precolumn, GRAM 100 and GRAM 1000 columns (PSS, Mainz, Germany) for measurement of longer chain distributions [46,47]. To obtain the chain length distributions (CLDs) of debranched starch molecules, the starch samples (5 mg) were debranched using isoamylase (I5284; Sigma, Louis, USA) before analysis.

### 4.6. Statistical Analysis

All data were represented as mean ± SD from at least three independent experiments with three replicates each. Significant differences between treatments were analyzed with one-way analysis of variance (ANOVA) and differences in means were compared using Duncan’s multiple range test. Statistical analysis was performed using SPSS Statistics 17.0 software (IBM Corp, Armonk, NY, USA). A value of *p* < 0.05 was considered a statistically significant difference.

## Figures and Tables

**Figure 1 ijms-20-04702-f001:**
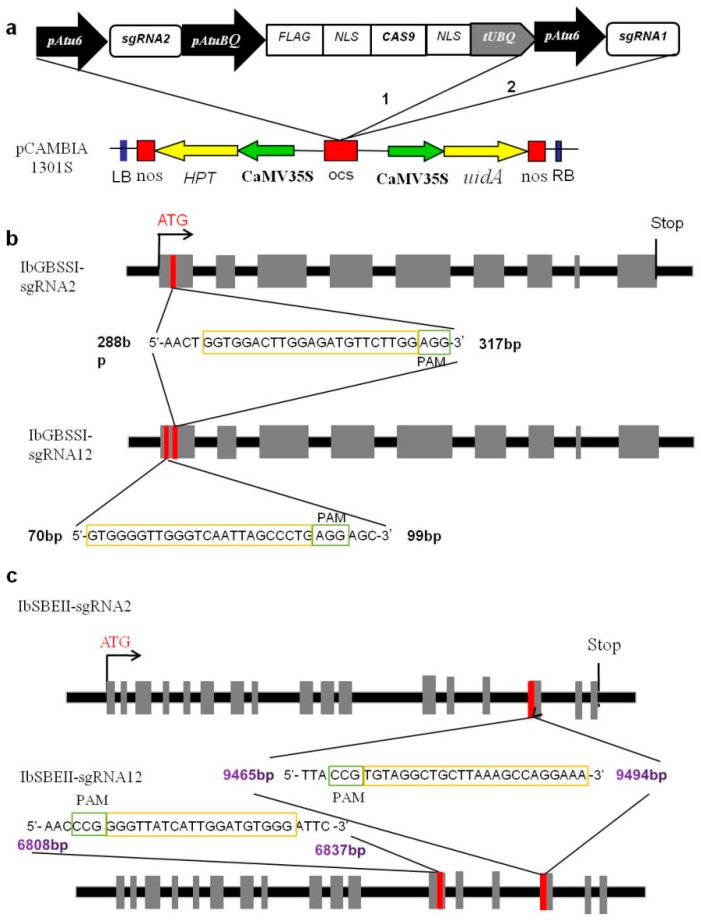
Schematic representations of the sweet potato granule-bound starch synthase I (*IbGBSSI*), and starch branching enzyme II (*IbSBEII*) target genes, location of the gRNAs, and CRISPR/Cas9 gene-editing construct. (**a**) Structural organization of the CRISPR/Cas9 binary vector pCAMBIA1301s used for stable *Agrobacterium*-mediated transformation in the sweet potato. *Arabidopsis thaliana* promoter AtU6 drives expression of each gRNA. The cauliflower mosaic virus promoter (CaMV 35S) drives expression of the *Cas9* gene. Abbreviations: 1, single gRNA vector; 2, double gRNAs vector; NLS, nuclear localization signal; Nos, Nos terminator. (**b**) Schematic representation of encoding granule-bound starch synthase I (IbGBSSI) target region and location of the gRNAs. Exons are shown as square frames and surrounding introns appear as lines. sgRNA and PAM are highlighted in yellow and green, respectively. (**c**) Schematic representation of IbSBEII target region and location of the gRNAs. Exons shown as square frames and surrounding introns appear as lines. sgRNA and PAM are highlighted in yellow and green, respectively.

**Figure 2 ijms-20-04702-f002:**
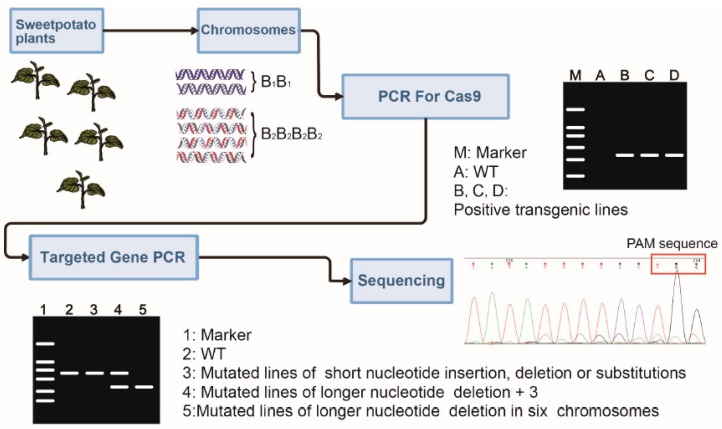
Schematic representation of the workflow designed to analyze targeted gene mutations of CRISPR/Cas9 editing. Transgenic lines were identified by PCR detection of Cas9 genes. Mutation detection in transgenic lines by PCR amplification with primers flanking the sgRNA target sites and running gel electrophoresis to roughly estimate the mutation types. PCR products sequencing analysis was performed by examining their sequencing chromatograms for accurate mutation status.

**Figure 3 ijms-20-04702-f003:**
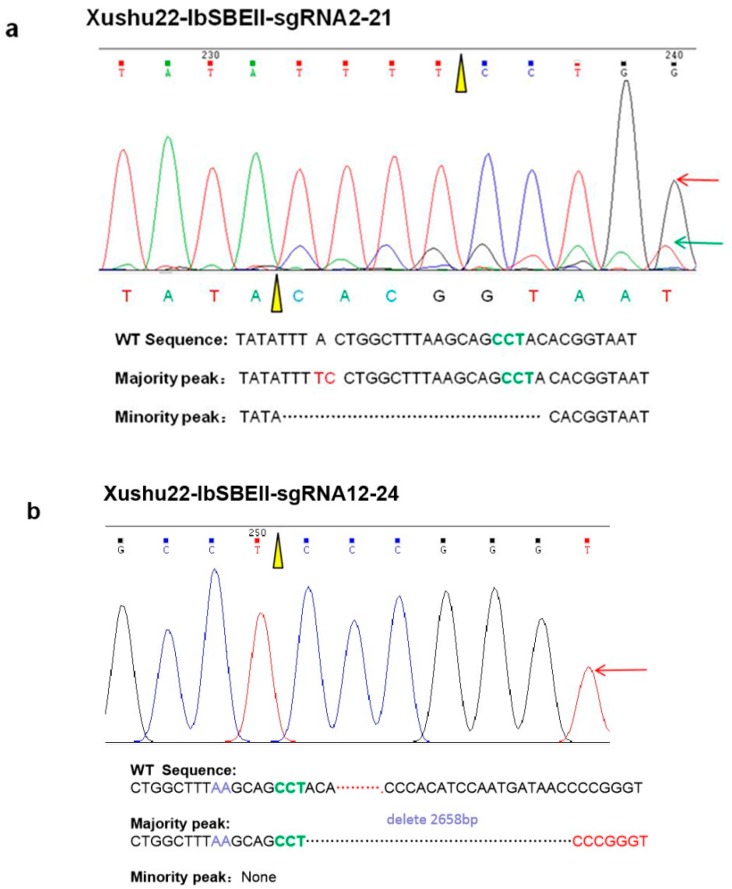
The typical sequence chromatograms of CRISPR/Cas9 editing produced targeted gene mutations. (**a**) Xushu22-IbSBEII-sgRNA2-21 multiple peaks appearance. Majority peak TC replace of A in wildtype sequence, while minority peak has a short sequence deletion compared with WT sequence. Dots indicate deletion sequences. Orange triangles indicate the position of sgRNA target site. (**b**) Xushu22-IbSBEII-sgRNA12-24 has a 2658 bp sequence deletion compared with WT sequence. Black dots indicate deletion sequences. An orange triangle indicates the position of sgRNA target site. Red line: T; green line: A; blue line: C; black line: G.

**Figure 4 ijms-20-04702-f004:**
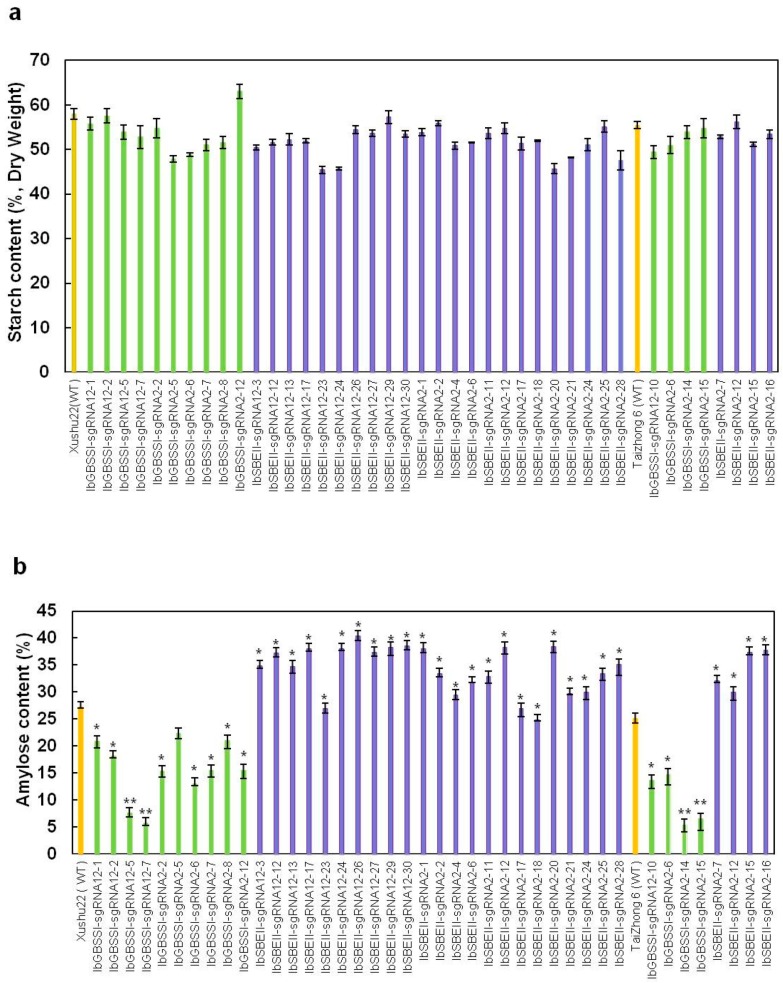
Total starch and amylose contents in transgenic lines produced using CRISPR/Cas9 systems. (**a**) Total starch content in storage roots of wild-types (Xushu22 and Taizhong6) and transgenic plant lines. (**b**) Amylose contents in storage roots of wild-types (Xushu22 and Taizhong6) and transgenic plant lines. Significance was determined by the Student’s t-test at * *p* < 0.05.

**Figure 5 ijms-20-04702-f005:**
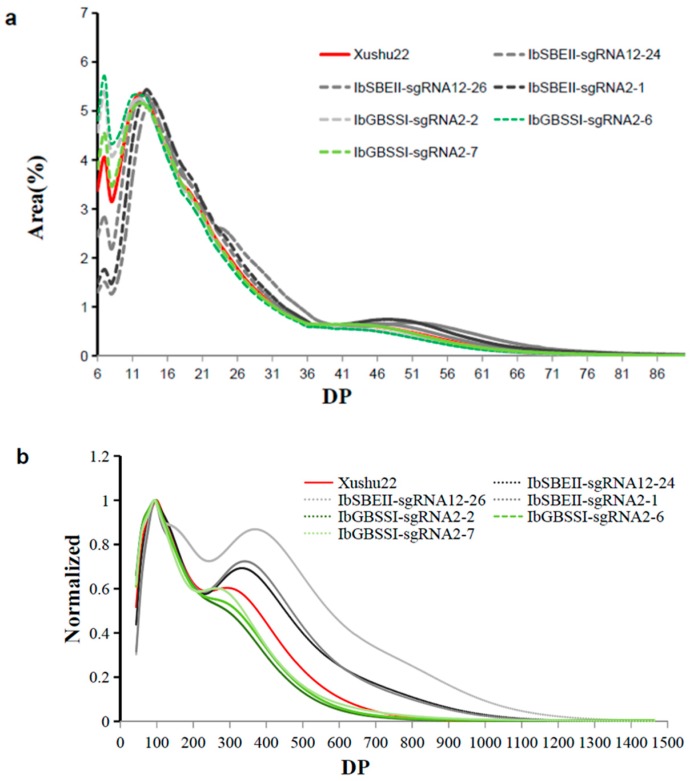
Chain length distributions of debranched sweet potato starches in transgenic lines. (**a**) Chain length distributions detection for the short chain of 90 > DP > 6 in Xushu22 and transgenic plant lines. (**b**) Chain length distributions detection for the long chain of DP >43 in Xushu22 and transgenic plant lines.

**Table 1 ijms-20-04702-t001:** Mutation rates of CRISPR/Cas9 in sweet potato.

Cultivar	Target Gene	No. of Plants with Cas9 Gene	No. of Plants with Mutation	Mutation Rates of Transgenic Plants
Xushu22	*IbGBSSI*	25	23	92.0%
Xushu22	*IbSBEII*	47	40	85.1%
Taizhong6	*IbGBSSI*	24	15	62.5%
Taizhong6	*IbSBEII*	11	7	63.6%

**Table 2 ijms-20-04702-t002:** Chain length distributions proportion in 90 > DP > 6 of debranched sweet potato starches^a,b^.

Sample	6 < DP < 12 (%)	13 < DP < 24 (%)	25 < DP < 36 (%)	37 < DP < 90 (%)
Xushu22	29.2 d (0.42)	42.7 c (0.752)	14.7 c (0.40)	13.4 c (0.35)
IbSBEII-sgRNA12-24	24.6 e (0.63)	44.6 b (0.53)	15.0 c (0.16)	15.8 b (0.78)
IbSBEII-sgRNA12-26	16.4 g (0.03)	45.6 ab (0.06)	19.2 a (0.04)	18.8 a (0.05)
IbSBEII-sgRNA2-1	19.1 f (0.25)	46.3 a (0.34)	16.3 b (0.04)	18.3 a (0.62)
IbGBSSI-sgRNA2-2	33.7 b (0.07)	41.7 cd (0.27)	13.8 d (0.13)	11.2 d (0.23)
IbGBSSI-sgRNA2-6	34.7 a (0.21)	41.4 d (0.46)	13.1 e (0.13)	10.8 d (0.31)
IbGBSSI-sgRNA2-7	30.4 c (0.08)	42.6 c (0.10)	13.6 d (0.23)	13.4 c (0.23)

a Standard deviations are given within parenthesis. b The values in the same column with different two letters (a and b, b and c, a and d, d and e, e and f, f and g) differ significantly (*p* < 0.05).

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
