# Peer review of "CRISPR/Cas9-Based Mutagenesis of Starch Biosynthetic Genes in Sweet Potato (Ipomoea Batatas) for the Improvement of Starch Quality"

_ijms, 2019, doi:10.3390/ijms20194702_

Round 1
Reviewer 1 Report
The authors used CRISPR/Cas9 technology to improve the starch quality of sweetpotato by targeting two critical starch biosynthetic genes, IbGBSSI and IbSBEII. Creating a mutation in the allopolyploid crop has been performed in wheat, this study could be the first case using CRISPR/Cas9 in sweetpotato. However, several questions still need to be addressed for further process of the manuscript.
1. In addition to the starch quality can be significantly improved by altering IbGBSSI and IbSBEII, how were other yield traits performed in these mutation lines?
2. To identify the mutations in the transgenic lines, the target site sequences were amplified using the flanking primers. However, the primers for SBEII (Table S2) could not amplify the second target sites of SBEII. It would be more clear to add the primer binding sites to Figure 1B and 1C.
3. To detect the mutations in the target genes, DNA sequencing of the PCR products was performed. The authors determined the mutation types by analyzing the sequence chromatograms. However, the intensity of majority peak and minority peak could be affected by sweetpotato 6x genomes, which could be difficult to determine the correct sequence.
4. In the starch quality assay (Fig. 4), several transgenic lines were omitted in Table S1, which including IbSBEII-sgRNA2-24, IbSBEII-sgRNA2-29, IbSBEII-sgRNA2-8.
5. According to the sequencing results, IbGBSSI-sgRNA2-7, IbSBEII-sgRNA2-12, IbSBEII-sgRNA2-17, IbSBEII-sgRNA2-18, IbSBEII-sgRNA2-20, and IbSBEII-sgRNA2-7 had identical sequence to wild type in the target genes. However, the amylose contents are significantly different than that in wild type. Authors should comment on this aspect.
6. No statistical analysis section in the material and methods. The statistical analysis method should be consistent throughout the manuscript. ANOVA should be done to provide statistical letters, indicating significant/insignificant among all genotypes.
7. In Figure 5b, the figure legends did not match with the graph.
Author Response
Responses to reviewers’ comments and suggestions
Reviewer #1:
The authors used CRISPR/Cas9 technology to improve the starch quality of sweetpotato by targeting two critical starch biosynthetic genes, IbGBSSI and IbSBEII. Creating a mutation in the allopolyploid crop has been performed in wheat, this study could be the first case using CRISPR/Cas9 in sweetpotato. However, several questions still need to be addressed for further process of the manuscript.
Re: We greatly appreciate the reviewer’s comments on the value of our study. We took into account all of the reviewer’s comments, suggestions and criticisms carefully and responded extensively, which helped us significantly improve the quality of our work.
In addition to the starch quality can be significantly improved by altering IbGBSSI and IbSBEII, how were other yield traits performed in these mutation lines?Re: Beside the starch quality, the yield of storage root was also recorded in different area for two years. Mutation lines for higher amylose reduced the yield compared to WT. However, higher amylose still has a higher economic value for special application. For example, high-amylose, degradation resistant starch (RS) can be specifically used as dietary fiber for controlling obesity and diabetes.
To identify the mutations in the transgenic lines, the target site sequences were amplified using the flanking primers. However, the primers for SBEII (Table S2) could not amplify the second target sites of SBEII. It would be more clear to add the primer binding sites to Figure 1B and 1C.Re: We greatly appreciate the reviewer’s comments. There has an error of number of primer position but the sequence of primer is right. We revised it that can amplify the second target sites of SBEII from +8367bp to +9730bp. We try our best to let this manuscript more clear.
To detect the mutations in the target genes, DNA sequencing of the PCR products was performed. The authors determined the mutation types by analyzing the sequence chromatograms. However, the intensity of majority peak and minority peak could be affected by sweetpotato 6x genomes, which could be difficult to determine the correct sequence.Re: you are right, It is a little difficulty to analyze the mutation of hexaploid
Sweetpotato. First of all, PAM Site and gel can help us to analyze the mutation type in Crispr/ Cas9 mutation. In additional, DNA sequencing of the targeted gene maybe show you the majority peak and minority peak. We checked every chemical base pairs in sequence by eyes compared to wild type sequence. However, DNA sequencing of the PCR products can show a majority peak but no minority peak using WT as template.
In the starch quality assay (Fig. 4), several transgenic lines were omitted in Table S1, which including IbSBEII-sgRNA2-24, IbSBEII-sgRNA2-29, IbSBEII-sgRNA2-8.Re: We are greatly grateful for the reviewer’s carefully detection. IbSBEII-sgRNA2-24 was wrongly named as IbSBEII-sgRNA2-23 in Table S1. We revised it in Table S1 as IbSBEII-sgRNA2-24. IbSBEII-sgRNA2-29 and IbSBEII-sgRNA2-8 were wrongly recorded. In fact, they are the same copy of IbSBEII-sgRNA2-25 and IbSBEII-sgRNA2-7, separately. Therefore the data of IbSBEII-sgRNA2-29, IbSBEII-sgRNA2-8 were deleted in Figure4.
According to the sequencing results, IbGBSSI-sgRNA2-7, IbSBEII-sgRNA2-12, IbSBEII-sgRNA2-17, IbSBEII-sgRNA2-18, IbSBEII-sgRNA2-20, and IbSBEII-sgRNA2-7 had identical sequence to wild type in the target genes. However, the amylose contents are significantly different than that in wild type. Authors should comment on this aspect.Re: We greatly appreciate the reviewer’s comments. IbGBSSI-sgRNA2-7, IbSBEII-sgRNA2-12, IbSBEII-sgRNA2-17,IbSBEII-sgRNA2-18, IbSBEII-sgRNA2-20, and IbSBEII-sgRNA2-7 are detected as transgenic lines by PCR amplify Cas9. However, Sequence analysis couldn’t be detected mutation while amylose content were significantly different than that in wild type. We speculated minor peak of low PCR product in those transgenic lines affected differentiate the mutation site. So we need to further detect this transgenic lines. We discuss this in Lines 180-183.
No statistical analysis section in the material and methods. The statistical analysis method should be consistent throughout the manuscript. ANOVA should be done to provide statistical letters, indicating significant/insignificant among all genotypes.Re: We greatly appreciate the reviewer’s comments. The statistical analysis method was added in the material and methods part in lines 303-308 .
In Figure 5b, the figure legends did not match with the graph.Re: We revised it for Figure 5b legends.
Reviewer 2 Report
Wang et al describe the mutagenesis of two gene of the starch synthesis pathway using the CRISPR/Cas9 method. The results presented here are very interesting and provide evidence of the suitability of this method to edit genes in Ipomoea batatas.
There is a question that I do not understand quite well and I would like to see clarified in the manuscript. I. batatas has 6 copies of each gene. The transformed plants analyzed have their six copies mutated? The transformed plants have both mutated and WT versions of the studied genes? The phenotype observed correspond to different dosage of functional genes? What is the method authors propose to obtain commercial lines with a specific trait modified using this technology?
Other minor points:
The expression “knockout” is usually employed for a loss-of-function mutation in a gene. However, it seems authors use this term for all kind of mutants. For instance, the subheading 2.2 (efficient gene knockout in sweetpotato) make reference to all the different mutants found. A title such as “efficient gene mutagenesis in sweetpotato” could describe better this paragraph.
Potato is a tuber, not a root. The expression in page 9 (line 221) should be corrected
Author Response
Responses to reviewers’ comments and suggestions
Reviewer #2:
Wang et al describe the mutagenesis of two gene of the starch synthesis pathway using the CRISPR/Cas9 method. The results presented here are very interesting and provide evidence of the suitability of this method to edit genes in Ipomoea batatas.
Re: We greatly appreciate the reviewer’s comments on the value of our study. We took into account all of the reviewer’s comments, suggestions and criticisms carefully and responded extensively, which helped us significantly improve the quality of our work.
There is a question that I do not understand quite well and I would like to see clarified in the manuscript. I. batatas has 6 copies of each gene. The transformed plants analyzed have their six copies mutated?
Re: We greatly appreciate the reviewer’s comments. In the manuscript, I. batatas is hexaploid. However, not all genes have six copies. It was reported at least two IbSBEI copies have been found in the sweetpotato through Southern hybridization[1]. In our manuscript, The mutation was analyzed by sequencing of PCR product. We can detect the mutation but don’t know the copies of mutated sites. We also want to know mutated site in six chromosomes. Sweetpotato has a composition of two B1 and four B2 component genomes (B1B1B2B2B2B2), and have one variant with 58 bp length genome sequence. According to Yang et al., (2017) method, three polymorphic sites with two alleles each would allow up to 23 = 8 haplotypes. Therefore, using random amplified polymorphic DNA can localize the mutation into one or several chromosomes of six chromosomes.
[1] Hamada, T., Kim, S.-H., & Shimada, T. (2006). Starch-branching Enzyme I Gene (IbSBEI) from Sweet Potato (Ipomoea batatas); Molecular Cloning and Expression Analysis. Biotechnology Letters, 28(16), 1255-1261. doi:10.1007/s10529-006-9083-x
[2] Yang, J.; Moeinzadeh, M.H.; Kuhl, H.; Helmuth, J.; Xiao, P.; Haas, S.; Liu, G.; Zheng, J.; Sun, Z.; Fan, W. et al. Haplotype-resolved sweet potato genome traces back its hexaploidization history. Nat .Plants .2017, 3, 696-703.
The transformed plants have both mutated and WT versions of the studied genes? The phenotype observed correspond to different dosage of functional genes?
Re: In this manuscript, IbGBSSI and IbSBEII were separately transformed into different WT( Xushu22 and Taizhong6). the IbGBSSI-knockout reduced, while the IbSBEII-knockout increased amylose percentage through detection of starch content. There are the similar phenotype with IbGBSSI-knockdown and IbSBEII-knockdown through RNAi interference [1,2].
[1] Kitahara, K.; Hamasuna, K.; Nozuma, K.; Otani, M.; Hamada, T.; Shimada, T.; Fujita, K.; Suganuma, T. Physicochemical properties of amylose-free and high-amylose starches from transgenic sweetpotato modified by RNA interference. Carbohyd. Polym. 2007, 69, 233–240.
[2] Shimada, T.; Otani, M.; Hamada, T.; Kim, S.H. Increase of amylose content of sweetpotato starch by RNA interference of the starch branching enzyme II gene (IbSBEII). Plant Biotechnol. 2006, 23, 85–90.
What is the method authors propose to obtain commercial lines with a specific trait modified using this technology?
Re: First of all, higher amylose and amylopectin product by Crispr/Cas9 technique which produce a huge economic value. In addition, IbGBSSI and IbSBEII were transformed into two cultivar for removal of the transgenes through crossing-breed which is a prerequisite for gaining regulatory approval for broaden commercial applications. T-DNA-free genome editing can potentially speed up breeding process in sweetpotato.
Other minor points:
The expression “knockout” is usually employed for a loss-of-function mutation in a gene. However, it seems authors use this term for all kind of mutants. For instance, the subheading 2.2 (efficient gene knockout in sweetpotato) make reference to all the different mutants found. A title such as “efficient gene mutagenesis in sweetpotato” could describe better this paragraph.
Re: We greatly appreciate the reviewer’s comments. We revised it in subheading 2.2 as “efficient gene mutagenesis in sweetpotato”.
Potato is a tuber, not a root. The expression in page 9 (line 221) should be corrected
Re: We greatly appreciate the reviewer’s comments. We revised it as a tuber.
Round 2
Reviewer 1 Report
The manuscript has been significantly improved.
Figures quality should be improved.
Figure 3. Please add the panel letter each sub-figure. The deletion size should be labeled as 2658 in the 3B.
Table 2. Please be consistent with the column label. The label of the last column should be "37<DP<90 (%)".
Author Response
V
Responses to reviewers’ comments and suggestions
Reviewer #1:
Re: The revised manuscript was reviewed and edited by a native English speaker. The major changes are tracked in the revised manuscript.
1 Figure 3. Please add the panel letter each sub-figure. The deletion size should be labeled as 2658 in the 3B.
Re: We are greatly grateful for the reviewer’s carefully detection. Panel letter a and b were added into Figure 3. We resubmitted Figure1, 3 ,4 as an independent file on September 4, 2019 according to reviewers’ suggestion. However, the word edition manuscript intercalated figure didn’t be updated. This time, the word edition manuscript intercalated Figure1, 3 ,4 were updated including 2658bp deletion size in Figure 3B.
2 Table 2. Please be consistent with the column label. The label of the last column should be "37<DP<90 (%)".
Re: We are greatly grateful for the reviewer’ suggestion. 90>DP>37 (%) was replaced of 37<DP<90 (%) for being consistent with the column label.